# Scanning Electron Microscopy Analyses of Dental Implant Abutments Debonded from Monolithic Zirconia Restorations Using Heat Treatment: An In Vitro Study

Marco Tallarico [1,*] , Łukasz Zadrożny [2] , Nino Squadrito [3], Leonardo Colella [4], Maurizio Gualandri [5], Daniele Montanari [3], Gianantonio Zibetti [6], Simone Santini [7], Witold Chromiński [8,*], Edoardo Baldoni [1], Silvio Mario Meloni [1], Aurea Immacolata Lumbau [1] and Milena Pisano [1]

1 Department of Medicine, Surgery, and Pharmacy, University of Sassari, 07021 Sassari, Italy
2 Department of Dental Propaedeutics and Prophylaxis, Faculty of Dental Medicine, Medical University of Warsaw, 02-091 Warsaw, Poland
3 Independent Research, 16100 Genova, Italy
4 Independent Research, 70126 Bari, Italy
5 Independent Research, 00100 Rome, Italy
6 Independent Research, 21052 Busto Arsizio, Italy
7 Independent Research, 53100 Soviclle, Italy
8 Faculty of Materials Science and Engineering, Warsaw University of Technology, 02-091 Warsaw, Poland
* Correspondence: me@studiomarcotallarico.it (M.T.); chrominski@pw.edu.pl (W.C.)

**Abstract:** Aim: The aim of this in vitro study is to present a debonding protocol developed to remove a screw-retained, monolithic, zirconia restoration from its titanium-base abutment, and to microscopically evaluate the abutment integrity at both the prosthetic and connection levels. Materials and Methods: A total of 30 samples were tested. Each sample consisted of a monolithic zirconia restoration bonded on a titanium link abutment. Five different shapes were designed and fabricated. Randomly, one-third of the Ti-link abutments were subjected to an anodizing process. Then, all the zirconia samples were bonded to the Ti-link abutments according to a pre-established protocol. Forty-eight hours later, the samples were debonded according to the experimental protocol. The outcomes were evaluated by a visual inspection with an optical microscope, scanning electron microscopy (SEM), and chemical composition analysis. Results: Thirty samples were collected and visually analyzed. Seven samples were randomly evaluated via scanning electron microscopy. In all the examinations, no relevant changes were reported. Chemical composition analysis also relieved no changes in the chemical structure of the titanium. Conclusions: The titanium-base abutments do not alter the structure and properties of the material, not creating phase changes or the birth of oxides such as to induce fragility. Further clinical studies with longer follow-up periods are needed to confirm these preliminary results.

**Keywords:** dental implant-abutment design; titanium; analysis; dental prosthesis retention; zirconium oxide

## 1. Introduction

Precision at the implant–abutment interface is one of the most important aspects influencing marginal bone remodeling and the risk of peri-implantitis [1]. Microgaps and bacterial leakage play an important role in peri-implant inflammatory reactions and the subsequent loss of supporting bone to restore the physiological biologic width [2]. Definitive abutment placed at implant insertion and not being removed seems to be an effective prosthetic approach to reduce the physiological marginal bone remodeling [3]. However, in recent decades, screw-retained implant restorations have increased in popularity due to their predictable retention, retrievability, and lack of potentially retained sub-gingival cement. The last point has become very important due to the trend to place implants subcrestally [4].

Prosthetically driven implant placement is crucial for the long-term success of treatment, allowing the implants to be installed in the most accurate mesio-distal and bucco-lingual position and depth [5–10]. Some studies show no clinical differences when placing implants 0.5 mm or 1.5 mm subcrestally; therefore, clinicians can choose as they prefer [11]. However, the depth of implant placement should be carefully planned to consider available bone and soft tissue thickness, the type of the implant, and the type and shape of further prosthetic reconstruction. Placing an implant in a subcrestal position may have a positive impact, especially in the aesthetic area, where obtaining a harmonious emergence profile is mandatory [9–11]. However, the vertical position mostly depends on the type of connection. Implants with internal conical connection and platform switching at the implant–abutment interface have been shown to maintain stable bone levels over a mean follow-up period of two years when placed subcrestally [12,13].

Due to their aesthetics, high mechanical properties, and biocompatibility, Yttria stabilized tetragonal zirconia ceramics have gained popularity as the preferred restorative material for implant-supported single crowns in the aesthetic area, with survival rates ranging between 90% and 96% after observation periods of 5 and 10 years, respectively [14–16]. For these reasons and more, implant companies market several prosthetic options to deliver screw-retained implant-supported restorations. Within these, titanium-base abutments (TBAs) or titanium-link abutments can be considered feasible treatment options for restoring dental implants [17]. The final restoration is a hybrid cemented-screwed, aesthetic solution composed of a metal-free restoration that is bonded outside of the patient's mouth to an original TBA [18]. The main benefits of this approach include its retrievability, highly precise implant–abutment fit (guaranteed by the manufacturer), and the customization of the emergence profile. Moreover, working on a fully or semi-digital workflow, hybrid prosthetic solutions also potential reductions in production costs compared to the classical workflow [19–22].

To create hybrid prosthetic solutions, monolithic zirconia or porcelain fused to zirconia (PFZ) restorations are computer-aided designed (CAD) and computer-aided manufactured (CAM) a with a semi-digital or fully digital approach [22]. Finally, the zirconia restorations are bonded chairside on TBAs, resulting in the hybrid cemented/screw-retained restorations. This approach reduces any inflammatory process due to cement remnants in the peri-implant tissue, maintaining its retrievability. Bonding can also be performed in a dental laboratory under controlled conditions; nevertheless, in case major ceramic corrections are needed (color, contact points, occlusion), the TBA must be debonded from the ceramic restoration before it is placed in the dental ceramic oven at 370 °C for five minutes. Moreover, when a zirconia restoration is debonded, the resin cement remains adhered to it, and this must be removed before the restoration can be re-cemented. In the literature, there are several papers concerning bonding protocols and retentive force [23–25]. However, at the time of writing this manuscript, and to the authors' knowledge, there are no manuscripts reporting debonding procedures and their impact on the surface of titanium abutments.

The aim of this in vitro study is to present a debonding protocol developed to remove a screw-retained, monolithic zirconia restoration from its TBA and to microscopically evaluate the abutment integrity at both the prosthetic and implant–abutment connection level.

## 2. Materials and Methods

A total of 30 samples were considered for this in vitro research. No similar study was found in the literature. For this reason, a priori sample size analysis was not performed. Each sample consisted of a monolithic zirconia restoration (MZR) bonded on a TBA (Ti-link Abutment, Osstem Implant, Seoul, South Korea). All the MZRs were designed (computer-aided design, CAD) and manufactured (computer-aided manufacturing, CAM) in one dental laboratory in Italy using a standardized protocol as recommended by the manufacturer (ST ML, UpCera Shenzhen Dental Technology Co. LTD., Nobil-Metal, Asti, Italy). The percent composition of the used zirconia was $ZrO_2$ + $HfO_2$ + $Y_2O_3$ > 98%;

Er$_2$O$_3$ < 1.0%; Fe$_2$O$_3$ < 0.3%; Pr$_2$O$_3$ < 0.2%; other oxides < 0.5%. Five different shapes were designed and fabricated, representing the possible extreme clinical variables (Table 1 and Figure 1). The main physical data are reported in Table 2.

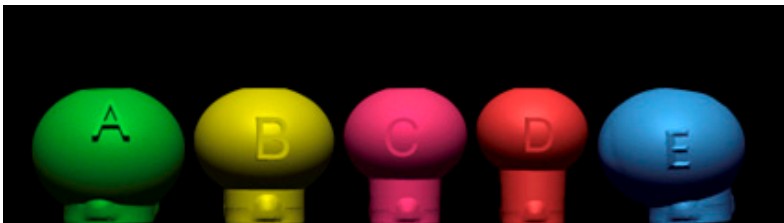

**Figure 1.** Computer-aided design (CAD) of computer-aided manufacturing (CAM) MZRs used as samples.

**Table 1.** Main characteristics of MZRs used in samples.

| Test Sample | Characteristics |
|---|---|
| A green | Standard anatomical center mass with 1 mm of gingival height |
| B yellow | Standard anatomical center mass with 2 mm of gingival height |
| C fucsia | Slightly shifted anatomical center mass with 3 mm of gingival height |
| D red | Standard anatomical center mass with 5 mm of gingival height |
| E blu | Shifted anatomical center mass with 2.5 mm of gingival height |

**Table 2.** Main physical data of the used zirconia.

| Physical Property | Data |
|---|---|
| Thermal expansion coefficients (CTE) | $10.5 \pm 1.0 \times 10^{-6}$ K$^{-1}$ (25–500 °C) |
| Density after sintering | $6.08 \pm 0.01$ g/cm$^3$ |
| Bending strength (three-point bending test) | >1200 Mpa |
| Sintering temperature | 1450–1580 °C |
| (recommended temperature) | (1530 °C) |
| Translucency | 43% |

Randomly, one-third (10 ut of 30) of the TBAs were subjected to an anodizing process in an anodizing bath heated at 20 °C with a solution of 10 g of trisodium phosphate (TPS) in 500 milliliters (mL) of distilled water under a current density of 5 mA $\times$ cm$^{-2}$ due to a stabilized anodizing potential of 65 V (titanium anodizer, Artiglio S.n.c., Parma, Italy). The anodizing process resulted in the formation of a gold-colored oxide layer with a thickness of about 120 μm in 30 s. Subsequently, all the zirconia samples were bonded to the TBSs according to a well-known protocol (Table 3), as follows.

**Table 3.** Bonding procedures.

| MZR | TBA |
|---|---|
| Sandblasting with aluminum dioxide 50 μm | Steam cleaning |
| Ultrasonic bath at low frequency for 10 min with distilled water | Drying with air |
| Drying with air | Primer * |
| Primer * | |

\* Clearfil Ceramic Primer Plus, Kuraray Noritake, Milan, Italy.

Teflon tape was used to seal the screw hole. Then, the MZRs were bonded to the TBA using PANAVIA SA resin cement (SA Cement Universal, Kuraray Noritake). An oxygen-inhibiting gel (Oxyguard II gel, Kuraray Noritake, Milan, Italy) was used to enable complete curing. Initially, a quick cure of 5 s was performed (Valo, Ultradent, Salt Lake City, UT, USA). After excess cement removal, the samples were put in a dental laboratory curing light and polymerized for 5 min. Finally, the samples were cleaned and polished (Figure 2).

Forty-eight hours later, the samples were debonded according to the experimental protocol reported in Table 4 and as previously published [23].

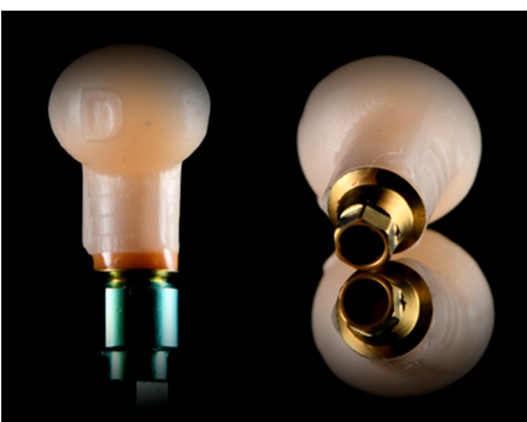

**Figure 2.** Titanium-base abutments after the bonding procedure.

**Table 4.** Debonding procedures: ceramic oven settings.

| **Settings and Procedures** |
| --- |
| After disinfection, remove the screw and place the restoration into the ceramic tray. |
| Pre-heating (initial temperature) to 300 °C. |
| Closing the oven for 2 min. |
| Stabilization for 5 min at 300 °C with the chamber closed. |
| Temperature increasing 30 °C per minute up to 370 °C. |
| Stabilization for 5 min at 370 °C. |
| Cooling for 2 min at 300 °C. |
| Opening the ceramic oven for 2 min. |
| Stabilization before handling for 10 min. |

The TBAs were removed from the zirconia restorations using a customized tool inserted inside the abutment screw access hole (Figures 3 and 4).

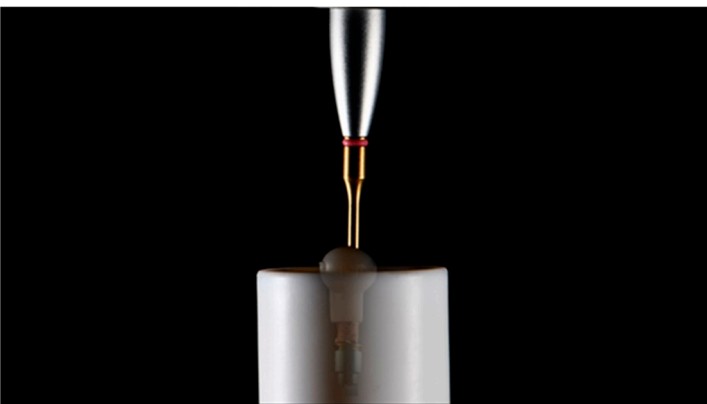

**Figure 3.** Customized tool used to remove the zirconia restorations.

Finally, all the TBAs were cleaned according to an established protocol (Table 5), inspected with the optical microscope, and analyzed via SEM.

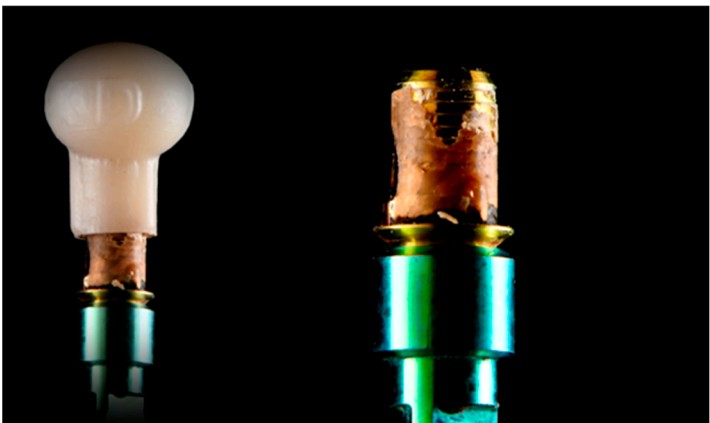

**Figure 4.** Titanium-base abutments immediately after the debonding procedure.

The SEM analyses were performed in two centers, one public centre in Warsaw (Warsaw University of Technology, Warsaw, Mazovia, Poland) and another private centre in Villafranca d'Asti, Italy (R&D Nobil Metal SpA). All the collected data were analyzed at the Department of Medicine, Surgery, and Pharmacy, University of Sassari, Italy.

**Table 5.** Cleaning procedure after debonding: treatment of the MZRs and the TBAs.

| MZR | TBA |
|---|---|
| Ultrasonic bath at low frequency for 1 min (distilled water). | Ultrasonic bath at low frequency for 15 min (distilled water). |
| Sandblasting with aluminum dioxide at 50 microns. | Ultrasonic bath at low frequency for 10 min with a liquid detergent (Pi-Ku-Plast Clear, Bredent). Manual cleaning. |

*Outcomes*

All the MZRs and TBAs were subject to a visual inspection with an optical microscope with different magnifications (up to 40× magnification value, Leica MS5 stereomicroscope, Leica, Milan, Italy) to evaluate the response of the zirconia to the applied debonding protocol, such as fracture and/or microscopic crack.

Randomly, 2 out of 10 anodized TBAs and one new titanium TBA (used as control) were examined via scanning electron microscopy (SEM) using a Zeiss EVO 10 SEM (R&D Nobil Metal SpA, Italy) operated at 20 kV to evaluate any kind of microscopic difference from the test sample.

Randomly, 5 out of 20 non-anodized TBAs (test) and one new TBA (used as control) were examined via scanning electron microscopy (SEM) using a Hitachi SU70 SEM operated at 30 kV to evaluate any kind of microscopic difference from the test sample.

Chemical composition analysis of all the analyzed samples (test and control TBAs) was performed by an EDS probe (Bruker—XFlash Detector, R&D Nobil Metal SpA, Italy) integrated into the Zeiss EVO 10 SEM (R&D Nobil Metal SpA, Italy).

## 3. Results

After the debonding procedure, all the MZRs and TBAs were visually inspected with a stereomicroscope. All the samples were debonded according to the aforementioned protocol. Then, all the MZRs were found to be free of complications, such as fractures or crack lines, independently of the shapes. Figure 5 shows a general view of the two TBAs—on the left side, the part is in an initial condition, and the right side shows the TBAs after the described procedures. The color difference between these two can be easily spotted. The initial condition maintains the typical outlook of the titanium, while the second became yellow. This is an expected outcome, as, during the heat exposure at 370 °C, an oxide layer is created on the titanium surface.

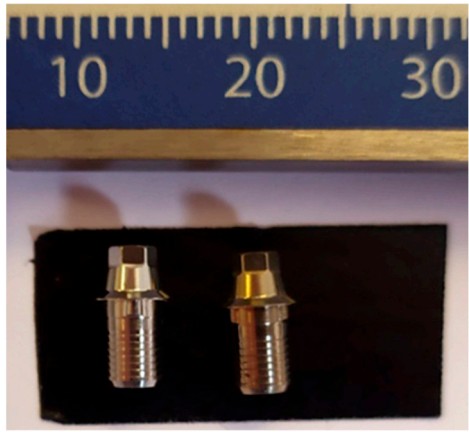

**Figure 5.** Control (**left**) and test (**right**) TBAs.

As no damages were found under stereomicroscopic analysis, the samples proceeded directly to the SEM inspection. More details can be found in the SEM images presented in Figure 6a–d. A conical part of the TBAs is compared in Figure 6a,b. As can be noticed, the contrast on the part after the thermal exposure is visibly less prominent, which may be related to a lower conductivity of the specimen or the presence of a very thin layer, which can hinder the escape of secondary electrons during observations. Both features can be linked to the presence of an oxide layer established during the thermal exposure of TBA. Higher magnification of the TBA in its initial condition reveals the patterns from the manufacturing process—the machining. The cementing and removal of the zirconia did not change those patterns, as seen in Figure 6d, but some slight changes on the surface can be noticed. Similar results were found for the anodized TBAs compared with the control and the new titanium-base abutment (Figure 7).

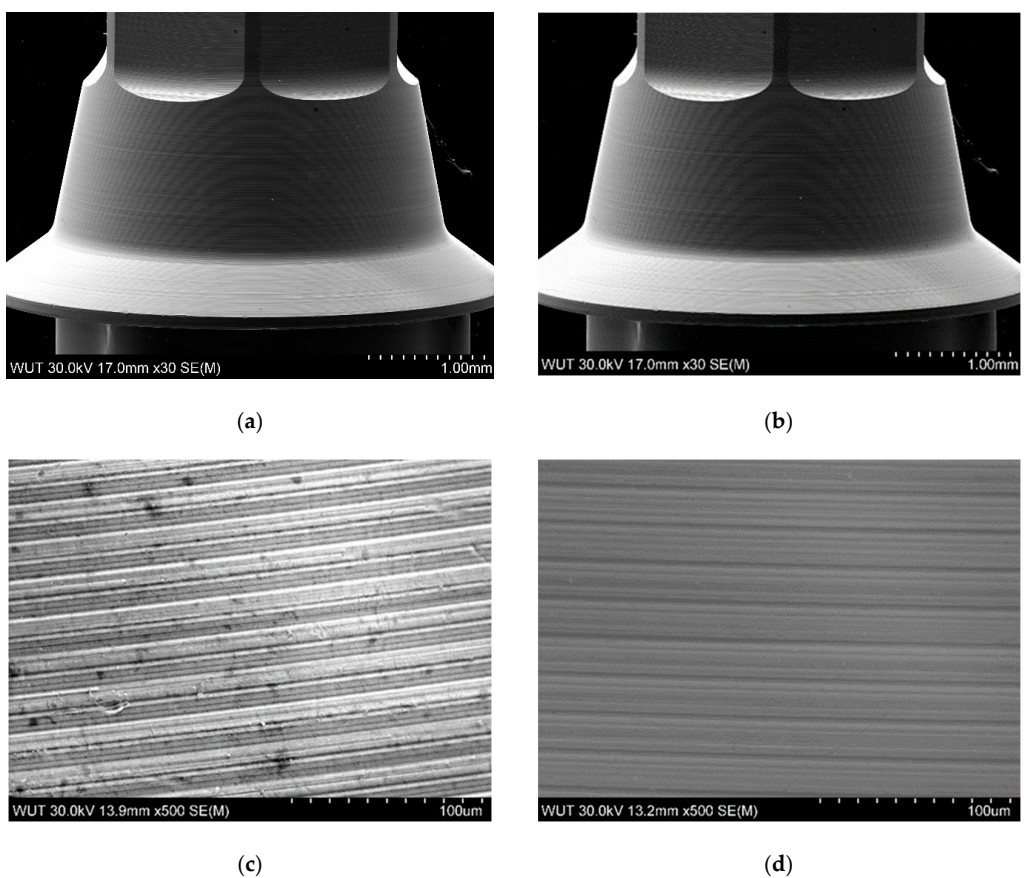

**Figure 6.** Test (**a**,**c**) and control (**b**,**d**) TBAs at 30× (**a**,**b**) and 500× (**c**,**d**).

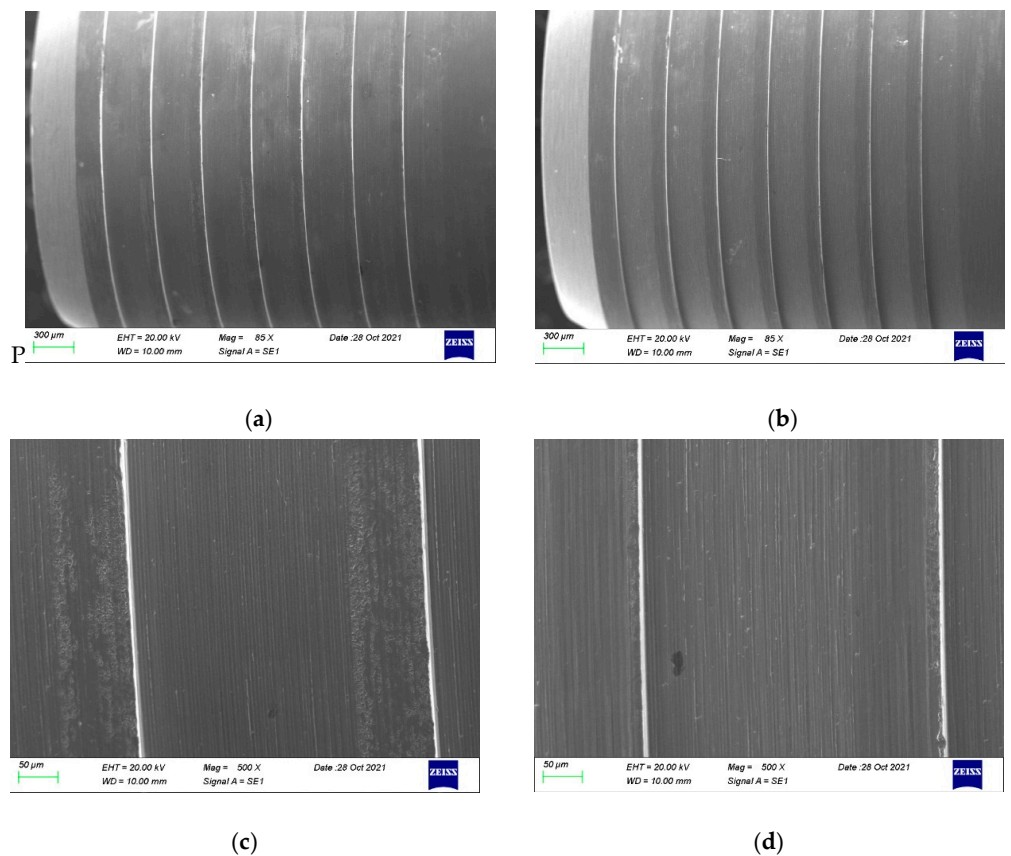

**Figure 7.** Anodized (**a**,**c**) and control (**b**,**d**) TBAs at 85× (**a**,**b**) and 300× (**c**,**d**). The black spot in Figure 7d is related to the electric charge in the dust on the specimen's surface.

The chemical composition analysis (Energy-Dispersive X-ray Spectroscopy [EDS] graph) showed no differences between the test and control groups (Figure 8).

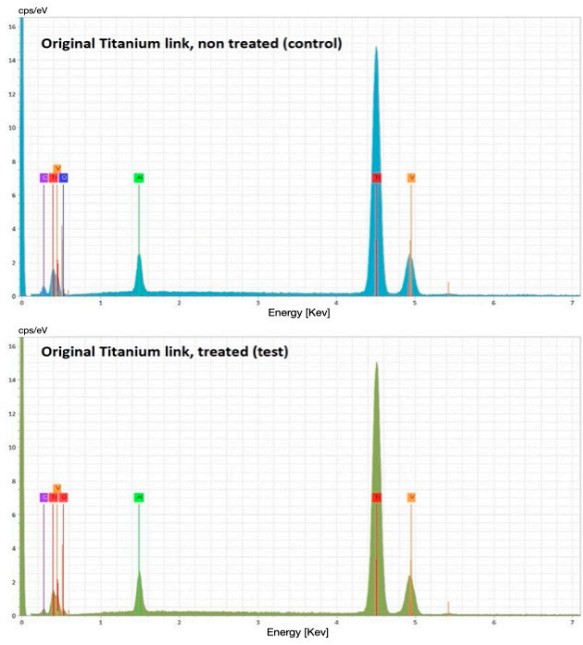

**Figure 8.** Energy-Dispersive X-ray Spectroscopy (EDS) graph (R&D Nobil Metal SpA, Italy).

## 4. Discussion

This in vitro study was developed to microscopically evaluate the effect of a debonding protocol applied to remove a screw-retained, monolithic zirconia restoration from its TBA. To the best of the authors' knowledge, at the time of writing this paper, there are no comparable studies. Therefore, it is impossible to compare the present research results with other studies.

One of the most important features of an implant-supported restoration is its retrievability, which could be necessary for implant complications. The monolithic ceramic restorations fulfill the need for suitable aesthetic reconstructions and reduce the risk of porcelain chipping. However, some complications, such as screw loosening, may still be observed [26–28]. In addition, interproximal contact could be lost at the implant sites, increasing the risk for periodontal disease [29]. The TBAs were introduced to overcome the risk of abutment fracture of one-piece zirconia restorations, allowing for a hybrid (cemented- and screw-retained), a strong link between the implants and the monolithic zirconia restorations, and finally providing a favorable, long-term, aesthetic outcome and patient satisfaction. Monolithic zirconia restorations bonded on TBAs can be easily retrieved by the patient's mouth; nevertheless, a strict protocol, such as the one presented in this research, must be applied.

In the present study, the analyzed samples were brought to a maximum temperature of 370 °C following standardized parameters avoiding the so-called "stress relief phase". This prevents structural changes usually obtained by titanium with a higher temperature. In the present study, stereographic microscope and SEM observations show the formation of oxides due to surface color changes to yellow (see Figure 6). From a material point of view, the process of debonding did not change the properties as well as the dimensions of the TBAs. The implant connection is free of any changes. Furthermore, the part where the crown was cemented had some minor but irrelevant changes.

Titanium oxide formation at high temperatures is a well-known fact, and it has been found to be influenced by the annealing temperature. Oxidation occurs due to the high reactivity of titanium with oxygen in the air, even at room temperature. The surface morphology and structural and electrical properties of $TiO_2$ films are influenced by the annealing temperature [30]. It was observed that when the annealing temperature increases up to 900 °C, the $TiO_2$ crystallite size is increased [30]. Nevertheless, at about 300° C, $TiO_2$ films crystallize in the anatase phase with poor crystallinity. In the same study, the calculated values of the crystallite size were less than 30 nm [30]. This means the connection part should be up to 30 nanometers bigger than the control TBA. Nevertheless, this is a transformative process that should not influence the overall dimension of the TBAs. Moreover, according to the literature, this might be clinically irrelevant [31,32]. It has been demonstrated that discrepancies greater than 10 μm result in microleakage and micromovements that allow for bacterial infiltration and mechanical outcomes, such as screw-loosening [18,30]. A sign of oxidation is discoloration due to a fragile layer enriched with oxygen near the surface (Alpha Case), which could be detrimental to the mechanical properties of the samples [33,34].

The main limitation of the present research is the in vitro nature of the study. Moreover, although 30 samples were fabricated and visually analyzed, only 7 out of 30 samples were randomly evaluated with SEM. However, all five randomly chosen samples showed the same results, thereby not justifying additional tests. In addition, the failure modes have not been registered, and no algorithm evaluation was conducted for statistical analysis. For these reasons, the data must be interpreted with caution. Nevertheless, the benefits of this research may be applied in several fields of implant dentistry, including prosthetic rehabilitation on implants [35–38].

Although no mechanical or clinical tests were performed, the main clinical considerations are that the TBAs could be reused in the same patient after debonding in those cases where the published protocol was applied. Nevertheless, the authors believe that the TBAs should be bonded chairside after clinical try-on, to avoid the need for debonding.

## 5. Conclusions

In light of what was observed from the SEM analysis, the treatment carried out on titanium-base abutments seemed not to alter the structure and properties of the material nor create phase changes or the birth of oxides to induce fragility. According to these results, titanium-base abutments may be reused after debonding. Further clinical studies with longer follow-up times are needed to confirm these preliminary results.

**Author Contributions:** Conceptualization, N.S.; M.T.; L.C.; M.G.; D.M.; G.Z.; S.S.; methodology, Ł.Z.; M.T.; N.S.; W.C.; writing—original draft preparation, Ł.Z.; M.T.; writing—review and editing, M.P.; S.M.M.; A.I.L.; E.B.; investigation, L.C.; M.G.; D.M.; G.Z.; S.S.; W.C.; supervision, M.P.; S.M.M.; A.I.L.; E.B. All authors have read and agreed to the published version of the manuscript.

**Funding:** This research received no external funding.

**Institutional Review Board Statement:** Not applicable for in vitro research.

**Informed Consent Statement:** Not applicable for in vitro research.

**Data Availability Statement:** Not applicable.

**Acknowledgments:** The authors thank the Vice-President and engineers, Christine Langenohl and Vincenzo Galati (R&D Nobil Metal Spa), and the Warsaw University of Technology that performed the SEM analyses.

**Conflicts of Interest:** Marco Tallarico, Łukasz Zadrożny, and SDT Nino Squadrito are key opinion leaders for Osstem Implant. Notwithstanding, all authors declare no conflicts of interest. OSSTEM AIC "Advanced Dental Implant Research & Education Center". Osstem AIC is a non-profit scientific community. Osstem AIC donated the titanium abutments used for this procedure; however, the data belongs to the authors, and by no means did the Osstem AIC interfere with the conduct of the study or the publication of its results.

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
