# Peer review of "Scanning Electron Microscopy Analyses of Dental Implant Abutments Debonded from Monolithic Zirconia Restorations Using Heat Treatment: An In Vitro Study"

_prosthesis, doi:10.3390/prosthesis4030041_

Round 1

Reviewer 1 Report (New Reviewer)

Dear Authors,

Your manuscript is really interesting and well conducted.

Unfortunately it cannot be published in present form and it needs to be revised.

1)Please be sure to use only keywords accordingly to medical subject headings (Mesh word) for a better indexing.

2)About the Title of the article, I suggest you to modify it and add the type of article

3) Also I suggest you add a table with the list of abbreviations used in the text.

4) I suggest to add more recent references about the topic of the article, dwelling in the introduction on articles published in 2022 and describing what your article will add compared to the last articles published; Preferably a published articles should be with 90 or more references, you don’t be afraid to add too many references.

Useful paper: Prosthodontic Treatment in Patients with Temporomandibular Disorders and Orofacial Pain and/or Bruxism: A Review of the Literature https://doi.org/ 10.3390/prosthesis4020025 

Telescopic overdenture on natural teeth: Prosthetic rehabilitation on OFD syndromic patient and a review on available literature PubMed ID 29460531

Can bone compaction improve primary implant stability? An in vitro comparative study with osseodensification technique DOI 10.3390/app10238623

5)Please expand conclusion section with main results and future perspectives of this study

Thank You,

Kind Regards

Author Response

Dear reviewer, thanks for your work

1)Please be sure to use only keywords accordingly to medical subject headings (Mesh word) for a better indexing.

- Key words have been changed accordingly.

2)About the Title of the article, I suggest you to modify it and add the type of article

- The study design has been added

3) Also I suggest you add a table with the list of abbreviations used in the text.

- Thanks for this suggestion. In general, MDPI doesn't include table with abbreviation. The abbreviation have been explained at the first time they in the manuscript.

4) I suggest to add more recent references about the topic of the article, dwelling in the introduction on articles published in 2022 and describing what your article will add compared to the last articles published; Preferably a published articles should be with 90 or more references, you don’t be afraid to add too many references.

Useful paper: Prosthodontic Treatment in Patients with Temporomandibular Disorders and Orofacial Pain and/or Bruxism: A Review of the Literature https://doi.org/ 10.3390/prosthesis4020025 

Telescopic overdenture on natural teeth: Prosthetic rehabilitation on OFD syndromic patient and a review on available literature PubMed ID 29460531

Can bone compaction improve primary implant stability? An in vitro comparative study with osseodensification technique DOI 10.3390/app10238623

- Dear reviewer, honestly I think 90 or more references are too much, nevertheless, I added the suggested references.

5)Please expand conclusion section with main results and future perspectives of this study

- Conclusions have been expanded with the main clinical consideration from this study.

Reviewer 2 Report (New Reviewer)

1.     The study design and purpose of this study need to be clarified more to avoid misunderstandings by readers.

2.     The study aimed to present a debonding protocol. Authors could compare the protocol with other protocols from the literature in the introduction and discussion.

3.      The study design needed to be clarified: (1)authors used five implant different shapes, but the authors didn’t analyze the results and discuss the deviation among these five types.

4.     Ten samples use anodizing process, and five different shapes were included in this study; the authors need to discuss the inter- and intra-group deviations.

5.     Please clarify the test group and control group; how many sample sizes are in it? And the authors’ study designed In materials and methods. Exp: Figure 5, left side is initial condition (Control group?) right side: yellow (test group?)

6.     Please correct e mistakes as follows:

7.     “3. Outcome” could include in material and method

8.     Please clarify Table 1 and Figure 1( clarify the “Decentralized”) 

9.     Line 137 (Figure 34?)

10.  Line 70 semi digital digital workflow (?)

11.  Line 226-228, Change to yellow (Figure 6?)

Author Response

Dear reviewer thanks for your precious work. Please check my responses.

  1. The study design and purpose of this study need to be clarified more to avoid misunderstandings by readers.

- Dear reviewer, the study design has been added in the title. Honestly i think the aim is clear for readers. 

  1. The study aimed to present a debonding protocol. Authors could compare the protocol with other protocols from the literature in the introduction and discussion.

- In the introduction is reported: "In the literature there are several papers concerning bonding protocols and retentive force (23-25), nevertheless, at the time of writing this manuscript, and to the authors' knowledge, there are no manuscript reporting debonding procedures and their impact over the titanium abutments surface."

  1. The study design needed to be clarified: (1)authors used five implant different shapes, but the authors didn’t analyze the results and discuss the deviation among these five types.

- Dear reviewer, this is a preliminary in-vitro study. Five different shapes were used to reproduce various clinical scenario. Statistical analysis were not performed in this preliminary research. In the discussion section these points have been discussed and limitations have been reported.

  1. Ten samples use anodizing process, and five different shapes were included in this study; the authors need to discuss the inter- and intra-group deviations.

- Dear reviewer, as previously reported, statistical analysis were not performed. In this preliminary report, the analyzed samples didn't allow statistical comparison. This limitations have been already reported.

  1. Please clarify the test group and control group; how many sample sizes are in it? And the authors’ study designed In materials and methods. Exp: Figure 5, left side is initial condition (Control group?) right side: yellow (test group?)

- A new titanium base abutment was used as visual control. This point has been clarified. Number of analyzed samples are already reported in the manuscript:

"All the MZRs and TBAs were subjected to a visual inspection with an optical microscope with different magnifications (up to 40x magnification value, Leica MS5 stereomicroscope, Leica, Milan, Italy) to evaluate the response of the zirconia to the applied debonding protocol, such as fracture and/or microscopic crack. Randomly, 2 out of 10 anodized TBAs and one new titanium TBA (used as control) were examined via scanning electron microscopy (SEM) using a Zeiss EVO 10 SEM (R&D Nobil Metal SpA, Italy) operated at 20kV to evaluate any kind of microscopic difference from the test sample. Randomly, 5 out of 20 non anodized TBAs (test) and one new TBA (used as control) were examined via scanning electron microscopy (SEM) using a Hitachi SU70 SEM operated at 30kV to evaluate any kind of microscopic difference from the test sample. Chemical composition analysis of all the analyzed samples (test and control TBAs), were performed by mean of EDS probe (Bruker – XFlash Detector, R&D Nobil Metal SpA, Italy) integrated into the Zeiss EVO 10 SEM (R&D Nobil Metal SpA, Italy)."

- Figure 5 has been fixed. Thanks a lot.

  1. Please correct e mistakes as follows:
  2. “3. Outcome” could include in material and method - DONE
  3. Please clarify Table 1 and Figure 1( clarify the “Decentralized”) - Centralized has been changed with shifted
  4. Line 137 (Figure 34?) - fixed
  5. Line 70 semi digital digital workflow (?) - fixed
  6. Line 226-228, Change to yellow (Figure 6?) - fixed

Round 2

Reviewer 2 Report (New Reviewer)

The authors have corrected the errors.

This manuscript is a resubmission of an earlier submission. The following is a list of the peer review reports and author responses from that submission.

Round 1

Author Response

Dear reviewer 1. Thanks for your suggestions.

- Please, in red my responses/corrections.

You wrote "The article starts from an interesting premise, however the presented analysis seem to be incomplete and the interpretation is lacking".

- Thanks again for any of your suggestions. This is an in-vitro study on a non previously analyzed topic. Authors focused on a debonding protocol and SEM analysis of link abutments. There is no guidelines for this procedure, very commonly used. The intent of this research is to be a preliminary report for further clinical research. I spotted a bit the conclusion where limitations are already reported. If you suggest why the presented analysis seem to be incomplete, I can add as limitation in the discussion. Thanks.

- English language has been revisited.

- °C has been used instead of Celsius degrees.

Table 1 - The color description used for the MZRs does not correspond to the CAD colors present in Figure 1. Also please use capital letters for sample designation, since the samples were fabricated wit capital letters embedded on the surface.

- Thanks a lot. The table has been fixed.

"For the electrolyte used in the anodizing process there is no mention of the solvent used or of the concentration of TBS." "and stabilized current of 65 V"

- Thanks a lot. Solvent and concentration  has been added, and current fixed.

- µm has been used instead of microns.

- Control and test in figure 5 have been identified.

- Captions of figures 3 and 4 have been modified accordingly.

- "Table 5. I suspect that is a cleaning procedure performed after the debonding?" Yes, it is the cleaning procedure. The text has been corrected. Thanks.

- Figures have been revisited. Thanks.

- Figure 6 has been changed (test and control has been inverted as suggested). Thanks a lot.

- The line in the discussed figure is related to electric charge in the dust on the specimen surface. We decided to exchange this figure with a new one, free from artifacts.

- Figure 7. Please provide a high resolution image, since the one in the manuscript cannot be read. Unfortunately I have only this. I defer to the editorial office. In case, I will delete it.

- Thanks for your precious comments. The discussion section has been modified.

"Titanium oxide formation at high temperatures is a well known fact and it have been found to be influenced by the annealing temperature. Oxidation occurs due to the high reactivity of titanium with oxygen, even at room temperature in air. The surface morphology, structural and electrical properties of TiO2 films have been found to be influenced by the annealing temperature (33). It was observed that when the annealing temperature increases from to 900°C, the TiO2 crystallite size is increased (33). Nevertheless, at about the 300 °C, TiO2 films crystallize in anatase phase with poor crystallinity. In the same study, the calculated values of the crystallite size was less than 30 nm (33). This means that the connection part should be up to 30 nanometers bigger than the control TBA. Nevertheless, this is a transformative process that ghouls not be influenced the overall dimension of the TBAs. Moreover, according to the literature, this might be clinically irrelevant (34,35). Has been demonstrated that discrepancies greater than 10 µm result in microleakage and micromovements that allow to bacterial infiltration and mechanical outcomes, such as screw loosening (18,32)."

- "The presence of an important layer could imply a fragility of the alloy." has been removed.

- A figure with anodized TBAs has been added.

Reviewer 2 Report

July, 5, 2022

Dear authors

Thank you for an interesting report.

In this study, you microscopically examined the abutment integrity at both prosthetic and connection level as for a debonding protocol developed to remove a screw-retained, monolithic, zirconia restoration from its titanium base abutments. In recent years, interest in digital dentistry has increased. Considering the dental practice in recent years, zirconia as a superstructure of titanium implants is in high demand, so this study is considered to be interesting. Therefore, it is thought that this research report will be useful for readers of dental field.

I agree to many parts of your claims and guessed that the subject of this paper will be of interest to the readership of Materials. However, I think that minor revisions are required as follows:

Page 3 Figure 1 and Table 1

1. It is difficult to understand the meaning of colors of each sample and the meanings of the markings A, B, C, D, and E in Figure 1.

2. It is also difficult to understand the meaning of a, b, c, d, e in Table 1 and the colors after the letters. I can't understand those mean in this article.

I think that you should explain them.

Page 3  Table 2

The table has not been created properly. You should make a table organized in two columns, physical characteristics and numerical values. Also, you should formally write the physical characteristic name, not the CTE.

Page 4  Tables 3 and 4

As for bonding/debonding procedures and ceramic oven settings, even if it is expressed in a table, it is difficult to understand easily. I think that it is easier for the reader to understand them if expressed in diagrams that show the operation flow at a glance.

Page 6-  Results

You randomly selected seven of the 30 samples and investigated, but only explained that there was no change and showed the results for the control and a tested sample. I think you should show the results of other 6 specimens in Figures 5 and 6.

Author Response

Thank you for an interesting report.

In this study, you microscopically examined the abutment integrity at both prosthetic and connection level as for a debonding protocol developed to remove a screw-retained, monolithic, zirconia restoration from its titanium base abutments. In recent years, interest in digital dentistry has increased. Considering the dental practice in recent years, zirconia as a superstructure of titanium implants is in high demand, so this study is considered to be interesting. Therefore, it is thought that this research report will be useful for readers of dental field.

Thanks a lot.

I agree to many parts of your claims and guessed that the subject of this paper will be of interest to the readership of Materials. However, I think that minor revisions are required as follows:

Page 3 Figure 1 and Table 1

  1. It is difficult to understand the meaning of colors of each sample and the meanings of the markings A, B, C, D, and E in Figure 1.
  2. It is also difficult to understand the meaning of a, b, c, d, e in Table 1 and the colors after the letters. I can't understand those mean in this article.

Figure and text have been modified also according to the reviewer 1. Thanks.

I think that you should explain them.

Page 3  Table 2

The table has not been created properly. You should make a table organized in two columns, physical characteristics and numerical values. Also, you should formally write the physical characteristic name, not the CTE.

Table has been modified as requested.

Page 4  Tables 3 and 4

As for bonding/debonding procedures and ceramic oven settings, even if it is expressed in a table, it is difficult to understand easily. I think that it is easier for the reader to understand them if expressed in diagrams that show the operation flow at a glance.

Thanks for this suggestion. Dental technician told me that this is an easier protocol to apply. I would prefer to leave like this. If also you agree. I also defer to the editorial office.

Page 6-  Results

You randomly selected seven of the 30 samples and investigated, but only explained that there was no change and showed the results for the control and a tested sample. I think you should show the results of other 6 specimens in Figures 5 and 6.

Thanks again. We presented some representative figures of the samples. A new figure has been added showing anodized samples compared with the control. I think now could be fine. Thanks.

Reviewer 3 Report

The authors cite articles that discuss clinical problems of zirconia with up to 10 years of clinical use as justification for this work. 

They propose the use of different specimens, but do not present the results for each restoration model. As there were 6 zirconia specimens, and TBA treatment in 10 of the 30 specimens, there is no balanced application of the zirconia specimens in each treatment.

There is no evaluation of cyclic loading which could support the results obtained in the long term. 

Apparently this work is a pilot study, and is incomplete.

Author Response

The authors cite articles that discuss clinical problems of zirconia with up to 10 years of clinical use as justification for this work. 

They propose the use of different specimens, but do not present the results for each restoration model. As there were 6 zirconia specimens, and TBA treatment in 10 of the 30 specimens, there is no balanced application of the zirconia specimens in each treatment.

Dear reviewer, thanks a lot. I agree that this is a pilot SEM analysis, and limitations have been discussed. Thanks.

There is no evaluation of cyclic loading which could support the results obtained in the long term. 

Dear reviewer, this is a SEM analysis. Of course, this is a pilot study for further research.

Apparently this work is a pilot study, and is incomplete.

I agree that this is a pilot study and it presents SEM and Chemical analysis. The limitations has been added in the discussion. Moreover, thanks to all the reviewers, the manuscript has been improved.

Reviewer 4 Report

The authors present an in vitro research work on the effects of a debonding protocol using heat treatment of monolithic zirconia restorations in the abutments.  The topic is very interesting and presents a clinically relevant question. The tested protocol is well described and the study model is in general adequate. Sample size calculation was not performed since no preliminary data was available, and this was adequately justified. However, there is a major flaw regarding the research strategy and results presentation, regarding the measurement of the considered outcomes. No quantitative analysis was presented, thus comparison between groups using statistical methods is not presented, and therefore the interpretation of these results is limited.

Regarding outcome 1, visual inspection under magnification, the prevalence of the failure mode should have been registered in each group (test/Control) so that it could be compared (n=10 in each group). Regarding outcome 2, SEM analysis, although it was qualitative, an algorithm of evaluation could have been set so that it could create a quantitative measurement to perform statistical analysis (n=2 for anodized and n=5 for non anodized). Regarding outcome 3 - Chemical composition, sample size is not clear, and semiquantitative analysis should have been used to better compare differences in chemicals composition.  Was it performed in all SEM analysed samples?

There are some other questions outline below.

- It is not clear what was the objective of testing different shapes and if the effect of the shape-related variables (gingival height, anatomical center mass position) on the final outcome was measured. An explanation  on the objective of this study design and data on the observed differences between different shapes should be provided. 

- What was the polishing protocol referred to in methods ("Finally, samples were cleaned and polished")? 

- The SEM analyses were performed in two centres. Was there any analysis of inter-centre agreement? What were the results?

- No mechanical tests were performed, and the effects of heat treatment on the mechanical resistance of the abutment was not assessed. 

Please review some typing errors:

"For these reasons and more, implant companies marketing several prosthetic options to delivery screw-retained implant-supported restorations.", consider using deliver, instead of delivery.  

"aatention", replace with atention

conclusion section heading should be separated from the main text. 

Author Response

The authors present an in vitro research work on the effects of a debonding protocol using heat treatment of monolithic zirconia restorations in the abutments.  The topic is very interesting and presents a clinically relevant question. The tested protocol is well described and the study model is in general adequate. Sample size calculation was not performed since no preliminary data was available, and this was adequately justified. However, there is a major flaw regarding the research strategy and results presentation, regarding the measurement of the considered outcomes. No quantitative analysis was presented, thus comparison between groups using statistical methods is not presented, and therefore the interpretation of these results is limited.

Dear reviewer, thanks to appreciate this research, and I agree with you. This is a SEM analysis only. Limitations of the research have been added and discussed. The conclusions have been smoothed.

Regarding outcome 1, visual inspection under magnification, the prevalence of the failure mode should have been registered in each group (test/Control) so that it could be compared (n=10 in each group). Regarding outcome 2, SEM analysis, although it was qualitative, an algorithm of evaluation could have been set so that it could create a quantitative measurement to perform statistical analysis (n=2 for anodized and n=5 for non anodized). Regarding outcome 3 - Chemical composition, sample size is not clear, and semiquantitative analysis should have been used to better compare differences in chemicals composition.  Was it performed in all SEM analysed samples?

Chemical composition has been explained and limitations of this search have been discussed accordingly your kindly suggestions. Thanks.

There are some other questions outline below.

- It is not clear what was the objective of testing different shapes and if the effect of the shape-related variables (gingival height, anatomical center mass position) on the final outcome was measured. An explanation  on the objective of this study design and data on the observed differences between different shapes should be provided. 

Five different shapes were designed and fabricated, in order to representing the most possible, extreme, clinical variables. Further data have been added. Thanks a lot.

- What was the polishing protocol referred to in methods ("Finally, samples were cleaned and polished")? 

This is reported in table 5. Thanks.

- The SEM analyses were performed in two centres. Was there any analysis of inter-centre agreement? What were the results?

One center analyzed the test and control, the other the anodized TBA and the control. No inter-center agreement was performed.

- No mechanical tests were performed, and the effects of heat treatment on the mechanical resistance of the abutment was not assessed.

- This point has been added in the discussion as limitation.

Please review some typing errors:

"For these reasons and more, implant companies marketing several prosthetic options to delivery screw-retained implant-supported restorations.", consider using deliver, instead of delivery.

- Done. thanks.

"aatention", replace with atention

- Thanks.

conclusion section heading should be separated from the main text.

- Done. Thanks. 

Round 2

Reviewer 1 Report

This is the second revision of the manuscript “Scanning electron microscopy analyses of dental implant abutments debonded from monolithic zirconia restorations using heat treatment”.

The article was improved and the majority of the initial comments were answered.

The issues that still remain:

Page 3. “stabilized current of 65 V”. You can use either “stabilized anodizing potential of 65 V” or simply “stabilized  voltage of  65 V”.

Table 3 micron should be either  “μm” or “micrometers”

Figure 8 (EDS graph) is still unreadable.

Please reread the article and correct the small typos.

I recommend the publication of this work in “Materials” journal after these minor revisions.

Author Response

Dear reviewer thanks for your second review and suggestions.

The article was improved and the majority of the initial comments were answered.

- Page 3. “stabilized current of 65 V” has been changed in “stabilized anodizing potential of 65 V” as suggested.

- Table 3 micron has been changed in “μm” as suggested

- Figure 8 (EDS graph) has been improved at the maximum.

- All the paper has been check once again for small typos. Thanks.

Reviewer 4 Report

The authors' corrections and comments were able to meet the original concerns. However, the final sentence in the discussion section : "the main clinical conclusions are that the TBAs can be reused in the same patient, after debonding, and in cases where the published protocol has been applied. Nevertheless, it is the author opinion that TBAs should be bonded in the dental clinic after all the clinical procedures, so as to reduce the probability of debonding. "  should be removed since it is not supported by the data (data is in vitro, and only evaluates one outcome even in vitro so to infer clinical applications is not possible). 

Author Response

Dear reviewer, thanks for the second review. I partially agree with you. I agree that all the conclusions must to be supported by the data. Nevertheless, I would like to discuss some clinical consideration, in the discussion section. If you agree, I have changed the sentence. Conclusions has been replaced with consideration, and all the sentence has been revisited. So that, this sentence is just a clinical consideration, and the conclusions are in agreement with you. Thanks a lot.

"Although no mechanical and clinical tests have been performed, the main clinical considerations are that the TBAs could be reused in the same patient, after debonding, in cases the published protocol was applied. Nevertheless, it is the author opinion that TBAs should be bonded chairside, after clinical try-in, so as to avoid the need for debonding."